# Electroplating and Ablative Laser Structuring of Elastomer Composites for Stretchable Multi-Layer and Multi-Material Electronic and Sensor Systems

**DOI:** 10.3390/mi12030255

**Published:** 2021-03-03

**Authors:** Simon P. Stier, Holger Böse

**Affiliations:** Center Smart Materials and Adaptive Systems, Fraunhofer Institute for Silicate Research ISC, 97082 Würzburg, Germany; holger.boese@isc.fraunhofer.de

**Keywords:** electroplating, conductive elastomers, composites, ablative laser structuring, stretchable electronics, wearables, sensors

## Abstract

In this work we present the concept of electroplated conductive elastomers and ablative multi-layer and multi-material laser-assisted manufacturing to enable a largely automated, computer-aided manufacturing process of stretchable electronics and sensors. Therefore, the layers (conductive and non-conductive elastomers as well as metal layers for contacting) are first coated over the entire surface (doctor blade coating and electroplating) and then selectively removed with a CO_2_ or a fiber laser. These steps are repeated several times to achieve a multi-layer-structured design. Is it not only possible to adjust and improve the work previously carried out manually, but also completely new concepts such as fine through-plating between the layers to enable much more compact structures become possible. In addition, metallized areas allow the direct soldering of electronic components and thus a direct connection between conventional and stretchable electronics. As an exemplary application, we have used the process for manufacturing a thin and surface solderable pressure sensor with a silicone foam dielectric and a stretchable circuit board.

## 1. Introduction

The availability of stretchable conductive materials is a key requirement for the further development of rigid circuit board-based electronics toward soft and wearable systems, not only when considering conducting paths themselves but also advanced components such as stretchable sensors and actuators. Despite the existence of intrinsic conductive polymers, filling non-conductive polymers with conductive particles remains a scalable and cost-effective way to gain conductivity for any polymer, including elastomers. This route has been used to develop, among others, highly stretchable composite electrodes consisting of polydimethylsiloxane (PDMS) and silver-coated copper flakes [1].

In contrast, highly integrated semiconductors, such as microprocessors, have still only been realized as rigid, or at best flexible, devices. In the medium term, stretchable electronic systems are therefore only conceivable as a hybrid composition of conventional semiconductors and intermediate elastic conductor paths. However, this composition requires good mechanical and electrical bonding between rigid and soft components. In addition to the integration of conventional components, multi-layer and precisely structured designs based on elastic materials are an important prerequisite for the realization of stretchable electronics and sensors, analogous to conventional printed circuit boards. This work is therefore dedicated to new approaches for both problems, the mechanical and electrical bonding between rigid and stretchable components with metallization as well as the structuring of elastic multi-layer systems, and tries to design a uniform and integrated process.

### 1.1. Metallization

To increase conductivity and improve solderability, in recent publications the method of physical vapor deposition (PVD) was used to coat non-conductive polymers with conductive material [2]. Among other things, this serves as a basis for subsequent electroplating [3]. The insufficient strength of the layer deposited via PVD is only compensated indirectly via mechanical interlocking of the electroplated layer with a porous substrate [4]. Alternatively, chemical processes have existed for quite a long time to metallize electrically non-conductive polymers [5]. However, this requires a complex, multi-stage process, partially using chemicals that are harmful to the environment and health.

An already conductive, filled polymer was not used as a substrate in any of the processes mentioned. The layer applied by PVD is also subject to high equipment costs. Moreover, an improvement in adhesion can only be achieved with additional structuring of the substrate, for example by creating a surface porosity. However, unlike electrodeposition, PVD cannot fill the undercuts of a porous surface due to its ballistic transport process, so anchoring of the deposited layer cannot be expected to the same degree. Chemical metallization also involves a high level of process effort and can only be used with certain polymers, in particular acrylonitrile-butadiene-styrene copolymer (ABS) [5].

Other approaches use processes selected from conventional printed circuit board technology, in which the usual rigid epoxy or flexible, but not stretchable polyimide matrix is replaced with a thermoplastic elastomer such as thermoplastic polyurethane (TPU), and the conductive tracks are designed in a meandering shape rather than in a straight line (cf. US2008257589A1 [6]). However, the cycle-resistant stretchability of this printed circuit boards is limited to only low strains. Schreivogel et al. [7] reported about 10,000 cycles at 5% strain and about 100 cycles at 20% strain. Gutruf et al. [8] reported over 100,000 cycles at 12% strain. Further publications reported that over 100% strain was achieved [9,10], but only for a few cycles. The fundamental problem is the high brittleness (compared to polymers) and the low tear propagation resistance of copper, so that under cyclic loading, rupture and total failure can occur at any time. Moreover, the thermoplastic elastomers used generally have higher elastic moduli and lower temperature resistances than, for example, silicone rubber.

As an alternative to the above-mentioned processes, the direct electroplating of an electrode consisting of a particle–elastomer composite is demonstrated in this work. Long-established processes are available for this purpose, both on the surface (immersion bath) and selectively (cf. tampon electroplating), using common metals (Cu, Ag, Au, Zn) and the appropriate electrolytes. Likewise, the full-surface metallization of a continuous elastomer electrode with subsequent structuring is possible using suitable subtractive methods such as ablative laser processing described below. Multi-layer structures with through-platings (vias) are also conceivable in this way.

### 1.2. Processing

#### 1.2.1. Conventional Manufacturing Processes

Conventional processing and structuring methods for processing liquid starting materials for silicone composites exhibit significant disadvantages. The structuring by film masks in combination with doctor blade or spray coating, which is frequently used on a laboratory scale, is, according to the subjective assessment of the authors, very labor-intensive, difficult to automate and offers only low reproducibility. At the same time, the process is technically limited in terms of minimum structure sizes (according to the experience of the authors not below 0.5 mm) and complexity of layer structures (especially regarding vias). Industrial processes such as screen printing or roll-to-roll processes, on the other hand, require expensive special tools for each specific structuring step. The exact design and parameterization of the process also depends heavily on the material properties and requires complex adjustment.

#### 1.2.2. Additive Manufacturing Processes

Additive manufacturing processes (3D printing) have therefore become established as an alternative. This eliminates the need for expensive tooling and is capable of automating the production. However, the specific (additive) manufacturing process is still highly dependent on the nature of the material. Common processes include:Selective Laser Sintering (SLS) for powdered thermoplastic polymers and metals.Fused Filament Fabrication (FFF) for thermoplastic polymers in filament formStereo Lithography (SL) for liquid resinsInkjet printing for low viscosity polymer solutions or suspensions

Dielectric elastomers are multi-layer material systems for actuator and sensor applications, which consist of alternating dielectric and conductive layers with thicknesses of approx. 100 μm are manufactured preferably with liquid silicone precursors. Especially for these material systems, there is no suitable additive manufacturing process that combines liquid, partly opaque starting materials, multicomponent printing and very fine layer thicknesses. SLS is not an option, since silicones are neither meltable nor sinterable. The same restrictions apply to extrusion via FFF.

The combination of extrusion and thermal or UV-induced curing, as developed by Wacker Chemie AG [11], is possible in principle, since multicomponent printing can be realized by several parallel print heads. However, the resolution is only 0.4 mm and the structure sizes are limited to >1 mm [12]. Thus, the required homogeneous film thicknesses of 100 μm and below are not feasible.

For SL processes, the layer thicknesses are feasible, but multicomponent printing can only be done with considerable effort using alternating resin baths, which is contrary to the objective of 3D printing. Inkjet processes, on the other hand, fail mainly due to the high viscosities (> 1000 mPa·s) of the silicone polymers and the relatively large filler particles (>1 μm) of the electrically conductive composites.

#### 1.2.3. Ablative Laser Structuring

To nevertheless enable largely automated, computer-aided manufacturing of multi-layer and multi-material systems, the concept of ablative laser-assisted manufacturing was evaluated in this work. In this process, the layers (conductive and non-conductive elastomers as well as metal layers for contacting) are first applied over the entire surface via doctor blading or electroplating and then selectively ablated with a laser. These steps are repeated several times to achieve a multi-layer-structured design. The coating process is completely separated from the structuring process, since the laser processing is always performed on the already cured material. This would also ensure transferability to other material classes such as solid silicones.

Laser structuring has already been used to manufacture single-layer sensors, actuators and circuit carriers [13,14]. However, this former published work usually involved additional steps [14] such as etching, bonding and transferring. In contrast, multi-layer structuring processes for elastic sensors and circuit boards in combination with electroplating, in which no additional processing steps are required apart from coating and laser structuring, are not yet known to the authors.

## 2. Materials and Methods

### 2.1. Materials

For the substrate layers and as matrix material for silicone foam layers, the commercial 2-component silicone RT 625 (Wacker, Munich, Germany) was used. Expandable hollow microspheres (FN100-SS, Matsumoto, Osaka, Japan) were used as blowing agents for the preparation of a compressible silicone foam for the dielectric pressure sensor. For the conductive silicone electrode, an in-house developed unreinforced silicone formulation *S2* (cf. Table 1) was filled with 20 vol% silver-plated copper flakes (eConduct 044000, ECKART, Hartenstein, Germany). A commercial acidic copper electrolyte (aqueous and sulfuric acid-based copper sulfate solution, Conrad Electronic SE, Hirschau, Germany) was used for electroplating.

### 2.2. Coating

#### 2.2.1. Substrate Layer

For the manufacturing of the substrate layer of RT 625, a mixture of liquid silicone precursors was prepared by mixing the components A and B of the commercial product RT 625 in the ratio 9:1 for 30 s at 2500 min^−1^ in a dual-asymmetric mixer (Speedmixer, Hauschild, Hamm, Germany). The silicone precursor mixture was then spread on a glass plate with a doctor blade with a velocity of 15 mm s^−1^. The gap width was set to 165% of the target layer thickness, e.g., 330 μm for a target layer thickness of 200 μm due to the known liquid film formation effect. Finally, the liquid silicone film was cured for 10 min using an IR radiator (approx. 100 °C).

#### 2.2.2. Conductive Silicone Composite Electrode Layer

Based on 1.0 g silicone formulation *S2*, 2.4 g metal flakes were added to obtain composites with 20 vol% particles. 1 g of volatile solvent (n-butyl acetate) was added to achieve a low viscous consistency. The mixture was homogenized by mixing with 3500 min^−1^ for 30 s. Subsequently, 16 mg of catalyst (platinum-divinyltetramethyldisiloxane complex, 0.3% Pt) was added, followed by a short mixing for 15 s at 3500 min^−1^.

The mixture was then doctor bladed with a velocity of 15 mm s^−1^ as an additional layer on the previously prepared layer(s) and cured for 10 using an IR radiator (approx. 100 °C), whereby the solvent evaporates. The gap width in reference to the surface of the previous layer was set to 200% of the target layer thickness, e.g., 200 μm for a target layer thickness of 100 μm, due to the shrinkage effect caused by the evaporation of the volatile solvent.

#### 2.2.3. Compressible Foam Layer

Based on 9.0 g A-component of RT 625, 1.0 g microspheres was added. The mixture was homogenized by mixing with 2500 min^−1^ for 5 min and expanded in the oven at 150 °C for 1 h. Subsequently, 2.5 g of solvent (n-Butyl acetate) and 1 g of catalyst-containing B-component were added, followed by a short mixing for 30 s at 3500 min^−1^.

The mixture was then doctor bladed with a velocity of 15 mm s^−1^ as an additional layer on the previously prepared layer(s) and cured for 10 min using an IR radiator (approx. 100 °C), whereby the solvent evaporates. The gap width in reference to the surface of the previous layer was set to 150% of the target layer thickness, e.g., 750 μm for a target layer thickness of 500 μm.

#### 2.2.4. Electroplating of Metal Layer

For galvanic metallization, the elastomer composite electrode with metal flakes was connected to the negative pole (cathode) of a DC voltage source and brought into contact with the electrolyte. Inside this electrolyte was the positive pole (anode) in the form of a copper sheet. With an applied voltage of about 0.7 V, the following redox reaction takes place:
Anode: Cu → Cu^2+^ + 2 e^−^
Cathode: Cu^2+^ + 2 e^−^ → Cu

Copper ions go into solution from the copper sheet anode while metallic copper is deposited on the elastomer cathode. The deposited copper mass *m_Cu_* is related to the current *I* applied over a period *t*, the molar mass *M_Cu_*, the elementary charge *e* and the Avogadro constant *N_A_* as follows:(1)$$m_{Cu}=\frac{I\cdot t\cdot M_{Cu}}{2\cdot e\cdot N_{A}}$$

The thickness $d_{Cu}$ of the deposited layer is given by the mass and the coated area *A*:(2)$$d_{Cu}=\frac{m_{Cu}}{\rho_{Cu}\cdot A}=\frac{I\cdot t\cdot M_{Cu}}{2\cdot e\cdot N_{A}\cdot \rho_{Cu}\cdot A}=k_{galv,Cu}\cdot \frac{I\cdot t}{A}$$

For copper, this results in *k_galv,Cu_* = 0.0369 mm^3^A^−1^s^−1^. After completion of electroplating, the coated elastomer electrode was removed from the electrolyte, cleaned with distilled water, and patted dry.

### 2.3. Microscopic Characterization of Electroplating Sample

To evaluate the electroplatability of silicone-metal composites, a two-layer test sample consisting of an approximately 350 μm thick silicone substrate and an approximately 100 μm thick electrode layer was prepared. Then, the electrode layer was coated with a current density of 0.01 A cm^−1^ for 10 min. According to Equation (Equation 2), the expected film thickness is about 2 μm.

To gain insight into the connection between the metal and the elastomer composite layers, scanning electron microscopic (SEM) images (ZEISS AURIGA 60, Carl Zeiss Microscopy, Oberkochen, Germany; 5 kV acceleration voltage, secondary electron detector and backscattered electron detector) were taken. For this purpose, a piece with dimensions of about 5 mm × 5 mm was cut from the prepared films. After embedding in epoxy resin, it was cut and then grinded and polished to obtain a side view of the metal and composite layer perpendicular to the doctor blading direction in the preparation step. The cut surface was subjected to an ion beam etching to allow for an artifact-free examination. Additionally, a layer with a few nanometers of carbon was sputtered onto the sample to avoid charging effects.

### 2.4. Ablative Laser Structuring

To evaluate the application of ablative laser patterning, the commercial dual-laser system (Epilog Fusion Pro 32, Epilog Laser, Golden, CO, USA) was used. The technical data of this system is presented in Table 2. The device has both a CO_2_ laser and a fiber laser. According to the manufacturer [16], the CO_2_ laser can cut and ablate all materials except metals, glass and ceramics with a wavelength of 10,640 nm. This includes the material classes of transparent silicones, silicone foams and opaque silicone composites used in this work. However, metallic layers cannot be processed due to reflection in the IR spectrum. This is possible with the fiber laser with a wavelength of 1062 nm. In addition, the opaque composite layers can also be processed with this laser, but not transparent and ceramic materials.

For a laser system with a single lens, the radius *w_f_* of the beam at the focal point is calculated according to Equation (Equation 3) with the wavelength *λ*, the focal length of the lens *f*, the beam quality *k* = 1/*M*^2^ with diffraction coefficient *M* and the radius of the beam in front of the lens *w_p_*.
(3)$$w_{f}=\frac{\lambda \cdot f}{\pi \cdot k\cdot w_{p}}$$

For the 3-inch lens used for both laser sources, using the manufacturer’s specifications for the 2- and 5-inch lens, the focal widths are 120 μm (CO_2_ laser) and 5.3 μm (fiber laser). However, due to the Gaussian shape of the intensity distribution of the beams, processing edges are not expected to be sharply defined but continuous.

### 2.5. System Manufacturing

The processing methods described above were applied to manufacture two different demonstrative devices. The first one is a stretchable circuit board with three LED, which demonstrates the combination of a stretchable system with stiff electronic components. The second test device concerns a dielectric elastomer pressure sensor as a structured multi-layer system with dielectric silicone foam and conductive silicone layers.

#### Stretchable Circuit Board

As a first demonstration of the described manufacturing procedures, the laser structuring of a simple stretchable circuit board, which supplies LED with power using a coin cell, was chosen. The schematic representation of the manufacturing process is shown in Figure 1 and the associated manufacturing parameters are depicted in Table 3. For the purpose of clarity, the schematic cross-sectional view shown in Figure 1a does not show the actual structure, but only three representative elements (a through-hole plating, an SMD connection and an expandable area in between). The top view in Figure 1b, in contrast, shows the actual layer geometries.

To fabricate a single-conductor-layer stretchable circuit board, the layer assembly can be fully fabricated prior to the structuring process (step 1.1–1.3). The layer assembly consists of a substrate layer (thickness approx. 100 μm), a particle-filled conductive silicone layer (approx. 100 μm) and an electroplated copper layer (approx. 2 μm).

Subsequently, the laser processing is carried out in three steps: First, the metal layer is removed with the fiber laser at the positions where a stretchable and conductive area should be located (step 2.1). In the next step, the fiber laser is also used to remove both the metal layer and the conductive elastomer layer, where a stretchable and electrically insulating area should be located or a drilled hole or cut must be made (step 2.2). Removal of the conductive elastomer layer with the fiber laser has the advantage over the CO_2_ laser that the ablation is automatically limited at the boundary between the opaque conductive elastomer layer and the transparent substrate layer due to the lack of absorption in the transparent layer. However, this area is post-processed with the CO_2_ laser at low power to remove a fine residual layer left by the fiber laser (step 2.3). Also using the CO_2_ laser, a hole is drilled (step 2.4) and the entire board is cut out (step 2.5). Electronic components can now be soldered onto the remaining metallized areas of the finished board (step 3.1). The parameters for laser processing were determined by a simple series of preliminary tests and visual inspection of the results.

For demonstration purposes, the board was equipped with an arrangement consisting of a series resistor (220 Ω), an LED and a miniature switch, all as surface-mounted devices (SMDs), in triplicate (LED blue, green, red) (see Figure 2). The circuit is powered by a button cell (3 V). The miniature switches connect the inner positive pole of the battery via one of the LEDs to the outer ground connection.

For an electromechanical characterization, another conductor strip was produced analogous to the connection between LED and pressure switch. Accordingly, this strip has metallic copper surfaces at both ends. Instead of the LED, a cable was soldered onto the 2 mm × 2 mm large surface. The conductor strip with cable was then inserted into a rheometer (MCR502, Anton Paar, Graz, Austria) equipped with a stretching tool (UXF12/EXT) and a resistance meter (for details on setup and evaluation, see Stier et al. [15]). The cable was clamped on the fixed element and the unoccupied copper area on the moving element, so that an applied strain stresses the solder joint. With this setup, approximately 5000 strain cycles between 0 and 50% strain were conducted while electrical resistance was recorded. Upon completion of the cycling, the sample was stretched to failure (tearing of the copper layer below the solder joint) and the force required to do so was recorded.

### 2.6. Pressure Sensor

The laser structuring of a simple dielectric pressure sensor with metallized contacting surfaces was chosen as a second demonstrative example. The fabrication was basically carried out with similar process steps as the manufacturing of the stretchable circuit board. The main difference is the integration of three electrode layers (thickness approx. 50 μm) and the use of two compressible dielectric layers (approx. 350 μm), in this case a foamed silicone. A profiled silicone film structure (such as the knob profiles used by Böse et al. [18,19,20]), on the other hand, would not easily allow the sensor to be manufactured without a bonding step. Thus, for the overall sensor manufacturing process (see Figure 3), a coating process (steps 1.1–1.7, see Section 2.2) was combined with a laser structuring process (steps 2.1–2.7, parameters see Table 4), with nine sensors being manufactured in parallel.

A substrate layer (step 1.1, thickness approx. 100 μm) was provided with a first electrode layer (step 1.2). The first electrode layer was circumferentially ablated with the CO_2_ laser (step 2.1) and covered with a first dielectric silicone foam layer (step 1.3) and a second electrode layer (step 1.4). The second electrode layer was also circumferentially ablated with the CO_2_ laser to lie within the orthogonal projection of the first electrode layer (step 2.2), and then covered with a second dielectric silicone foam layer (step 1.5). To contact the now interior first and second electrode layers, holes were drilled through each of the two dielectric foam layers with the CO_2_ laser (steps 2.3, 2.4). The subsequently applied third electrode layer fills the created cavities and is therefore initially connected to both the first and second electrode layers.

A full-surface metallic copper layer (thickness approx. 10 μm) is now electroplated onto the third electrode layer (step 1.7). The metallized third electrode is divided into two sections with the fiber laser (step 2.5) and a subsequent post-processing with the CO_2_ laser (step 2.6): an inner ring connected to the inner, second electrode and an outer ring connected to the bottom, first electrode. Subsequently, the sensors are separated with the CO_2_ laser (step 2.7) and contacted with the metallized side by solder connection (step 3.1).

### 2.7. Sensor Characterization

To characterize the fabricated sensor, a force or pressure-capacitance curve was generated. For this purpose, the sensor was placed in a rheometer (MCR501, Anton Paar, Graz, Austria) equipped with a pressure measuring plunger and subjected to an increasing normal force up to about 8 N (corresponding to 25 kPa). The capacitance between the outer and inner electrodes was recorded continuously. After reaching the maximum force, a hold time of 30 s was applied to determine sensor creep. The sensor was then unloaded again and another hold time of 30 s was applied to record a full hysteresis loop.

## 3. Results

### 3.1. Electroplating Sample

In the cross-sectional view of the SEM images, the silicone electrode with the flat copper flakes (top) and the approximately 2 μm thick metallic copper layer (bottom) are clearly visible (see Figure 4). Here, the copper layer has grown directly onto the particles protruding from the electrode surface. The near-surface pore on the right side is also completely filled with metallic copper. Overall, this connection results in a very good mechanical interlocking of the metallic layer with the conductive silicone layer. In addition, the choice of suitable metals, such as copper in this case, results in smooth surfaces that are easy to solder. When using heat-resistant elastomers, such as silicones, soldering process temperatures of over 200 °C are thus possible without material degeneration.

### 3.2. Stretchable Circuit Board

With the specified manufacturing parameters, it was possible to produce the elastic circuit carrier as planned. In particular, it was possible to remove the electroplated copper layer in the stretchable areas without damaging the conductive silicone layer underneath. In addition, the conductive silicone layer could also be removed in insulating areas without damaging the substrate layer. A thin haze of the combustion residues remaining on the substrate is merely a visual impairment and not a functional one.

This result makes it possible to first apply both coatings over the entire surface of the substrate and then to work out any fine details with the laser. Conversely, it would be considerably more time-consuming to electroplate several individual areas directly, as this would require an individual tool for each geometry. Chemical etching of the electroplated copper layer is also conceivable, but would involve additional work steps (application and removal of photoresist), and the conductive silicone layer could not be removed in the same work step.

After soldering on all components (button cell, resistors, LEDs and switches), the demonstrative function of the circuit board was validated (see Figure 5a). Even in the stretched state, the adhesion of the components and the conductivity of the conductive tracks were sufficient (see Figure 5b).

### 3.3. Electromechanical Characterization

The electromechanical characterization of the laser-manufactured conductor strip yields an initial conductivity of about 2250 S m^−1^. The conductivity drops to 25% during stretching, but recovers to 75% when the strain is kept constant during the holding time (30 s) (see Figure 6a). In the first stretch cycle, the conductivity is slightly lower, but in the following cycles it remains largely unchanged. In the process of the durability test, the conductivity decreases significantly (see Figure 6b). In the first 100 cycles, the value of the conductivity in the stretched state is higher than the value in the unstretched state. This could be explained by the possible alignment effects of the particles [15]. Subsequently, the value in the stretched state is below the value in the unstretched state and both values continue to decrease. This behavior is not optimal, but corresponds to the previously reported behavior for the specific metal-flake-silicone composite prepared without laser patterning [15]. Thus, the degeneration is not due to the laser patterning or to any detachment of the metallic copper layer, but to the elastic composite itself. To provoke detachment of the copper layer, the specimen was finally stretched to failure. Only when the elastic part of the specimen was stretched by approx. 300% and a shear stress of 140 kPa was applied between the copper layer and the silicone composite, the connection failed (see Figure 6c). On the photograph of the fracture surface (see Figure 6d), which shows a remaining composite layer on both halves of the fracture, it can be seen that the failure does not occur directly between the metal and the composite, but within the composite layer.

### 3.4. Dielectric Elastomer Pressure Sensor

Totally nine sensors were fabricated in parallel from the multi-layer silicone composite film, three of which were coated with an electroplated copper layer (see Figure 7). One of these sensors is shown in detail in Figure 8.

As can be seen in the top view, the connection side of the sensor is covered with a copper layer. However, the inner and outer electrodes are cleanly separated from each other. Due to the curvature of the sensor when it is detached from the glass substrate, the copper layer is slightly corrugated, but not detached at any point. The cross-sectional view shows that the foam and electrode layers are largely homogeneous. The insulation ring between the inner and outer electrode is not visible in the cross-sectional view due to the electrode material behind it. However, the vias are not completely filled due to the volume shrinkage of the electrode material, caused by the evaporation of the volatile solvent in the preparation step. Even though an electrical contact is thus established, the mechanical stability could be affected by these voids. At this point, solvent-free formulations should be used in the future to avoid adverse shrinkage effects. With a thickness of approx. 1 mm, the sensor is relatively thin. In addition, the structure is closed all around, which is advantageous in an environment with liquid water.

### 3.5. Sensor Characteristics

For the functional validation of the sensor with a total area of 4 cm^2^, the force-capacitance curve was determined using a with the rheometer (see Figure 9).

The sensitivity measured with approx. 0.1 pF cm^−2^N^−1^ is relatively low and the hysteresis with approx. 20% is relatively high. Conventionally manufactured knob profile sensors achieve significantly better characteristic values with sensitivities of approx. 250 pF cm^−2^N^−1^ and hysterizes in the range of a few percent [19]. However, a laser-structured foam sensor is nevertheless suitable for applications in which the focus is not on high sensor quality but on simple and scalable manufacturing, especially since the smaller capacitive measuring range means that part of the low sensitivity can be compensated for by more accurate measurement on the part of the sensor electronics.

## 4. Discussion

The advantage of the electroplating process described in this work is the combination of a conductive particle–elastomer composite with a highly conductive, flexible as well as solderable metal layer. For this purpose, established and cost-effective processes such as electroplating baths or pad electroplating can be used and combined with the technology of elastic conductors. As the created SEM images show, the electroplated metal layer is firmly interlinked with the porous surface of the composite. The process thus creates a direct mechanical and electrical interface between the elastic conductor and conventional electronics.

With the high electrical conductivity of an electroplated flexible and solderable metal layer, a wide variety of applications can be realized. Through additional wave- [2] or meander-shaped [6] structuring, which has already been demonstrated for other substrates, the metal layers can also be made pseudo-stretchable. The high electrical conductivity enables the realization of flexible and elastic power electronics such as heating elements, coils, antennas or conductive tracks to supply energy-intensive components such as radio modules. Particularly relevant is the wireless data and energy transmission by inductive (cf. LF/HF-RFID or NFC) or electromagnetic transmission (UHF-RFID), e.g., for the use in wearable electronic systems.

As demonstrated in this work, the good mechanical bonding of the metallic layer to the conductive elastomer layer and its wettability by solder enable the bonding of conventional surface-mounted devices (SMDs) by established soldering processes. In addition to solderability, the metallized sections also allow for localized stiffening, which relieves the strain on the solder joints in the case of circuit carrier elongation. Instead, the strain is absorbed by the sections in which the elastic particle–elastomer composite is not metallized. In particular, capacitive or resistive elastomer sensors and elastomer actuators can be combined directly with the necessary control, evaluation and transmission electronics on an elastic circuit board. In the future, commercially available stiff semiconductor sensors such as temperature, light, humidity, acceleration and position sensors can also be connected with flexible and stretchable conductor paths to attach them to textile or generally moving surfaces (e.g., human skin, robot joints).

Although the particle–elastomer composite and the metal layer are reliably bonded, the fatigue strength of the particle–elastomer composites used is still limited. In particular, high strain rates and several thousand strain cycles significantly [15] reduce the conductivity of the material in the stretched state. In addition, the particle–elastomer composite used here is also the mechanical weak point, since the failure does not occur between the composite and the substrate or the composite and the electroplated copper layer, but within the composite layer. Although carbon black composites are more robust, they exhibit lower conductivity and are not suitable for electroplating. Further research on robust, highly conductive and elastic metallic elastomer composites is therefore a necessary basis for further development of the technology presented here.

The laser structuring of dielectric, conductive and metallized elastomer layers described in this work, offers considerable advantages over the manual structuring by using film masks and similar methods that have been common up to now, especially in combination with electroplating. Since the properties of the materials used (viscosity, particle size, polymer chemistry) do not have to be matched to the structuring process and no tools (masks, screens, molds) must be manufactured and positioned, development and manufacturing times can be significantly reduced. The seamless transfer of digital manufacturing data to the digital structuring process can support the progress of process digitization and eliminate the need for manual intermediate steps.

In addition, laser structuring also enables the realization of completely new concepts that cannot be implemented with mask structuring or printing technologies. These include, in particular, vias between individual electrodes in different layers. In combination with smaller dimensions and distances between the electrodes, much more compact layer structures can be produced. Thus, beyond the single-conductor-layer circuit board and simple pressure sensor demonstrated in this work, complex multi-layer circuit boards as well as spatially resolved and multimodal elastomer sensor systems (cf. Stier and Böse [21]) can be created in the future.

There is still room for improvement regarding the laser beam shape: The usual Gaussian profile limits the precision of machining, since machining edges always have a relatively large radius. A largely rectangular beam profile (Top hat), on the other hand, would allow smaller radii and thus step-shaped structures. Another disadvantage is that combustion residues remain on the machined surfaces. Here, effective cleaning methods, such as by compressed air or washing off, must be investigated. In addition, higher-energy lasers (such as UV lasers) could offer advantages.

In summary, the novel processes described enables the scalable and automated production of flexible and stretchable electronics with directly integrated passive and active stiff components as well as dielectric elastomer sensors. Potentially, the concepts can also be extended to other elastomer technologies such as magnetorheological or dielectric elastomer actuators, dielectric elastomer generators as well as resistive elastomer sensors.

## 5. Patents

Parts of the work leading to this paper have been patented under publication number EP3608922A1 [22].

## Figures and Tables

**Figure 1 micromachines-12-00255-f001:**
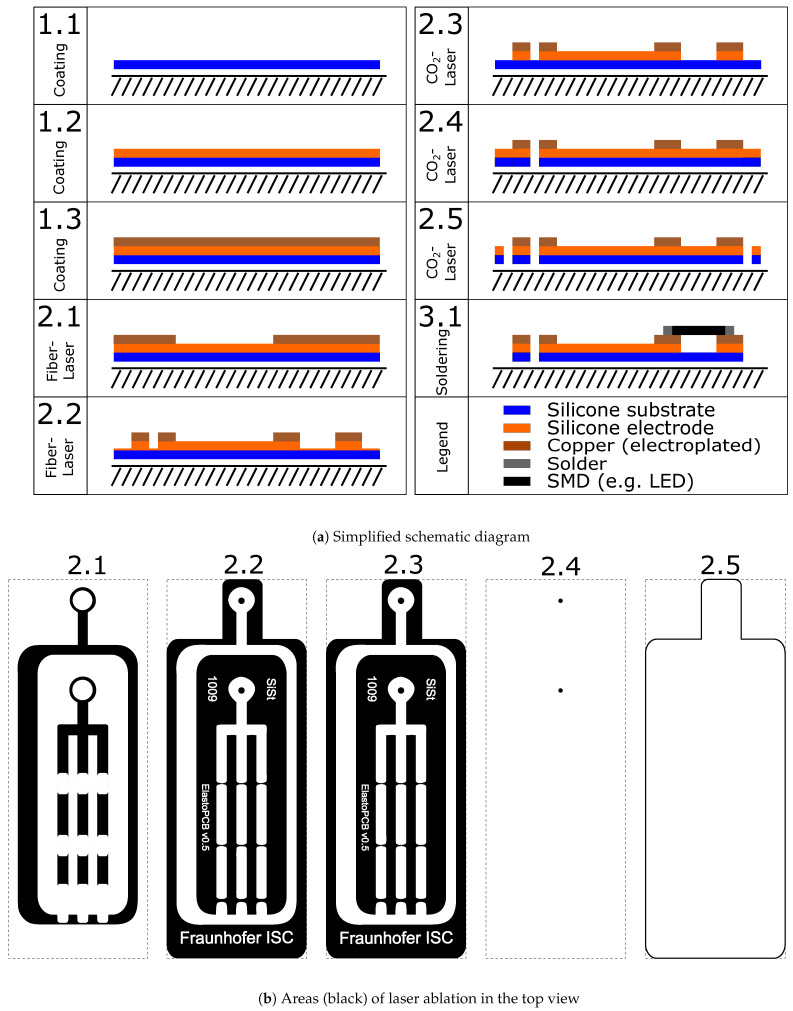
Simplified schematic diagram of the coating and laser structuring steps for manufacturing a single-conductor-layer circuit board (drawing not to scale). For the purpose of clarity, the schematic cross-sectional view shown in (**a**) does not show the actual structure, but only three representative elements (a through-hole plating, an SMD connection and an expandable area in between). The top view in (**b**), in contrast, shows the actual layer geometries.

**Figure 2 micromachines-12-00255-f002:**
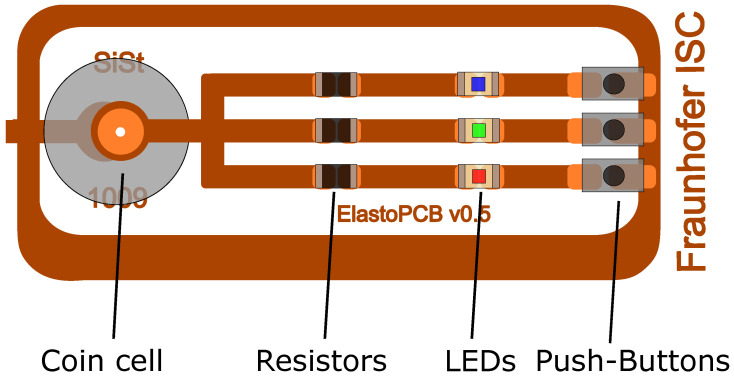
Schematic diagram of the structure and assembly of the stretchable circuit board.

**Figure 3 micromachines-12-00255-f003:**
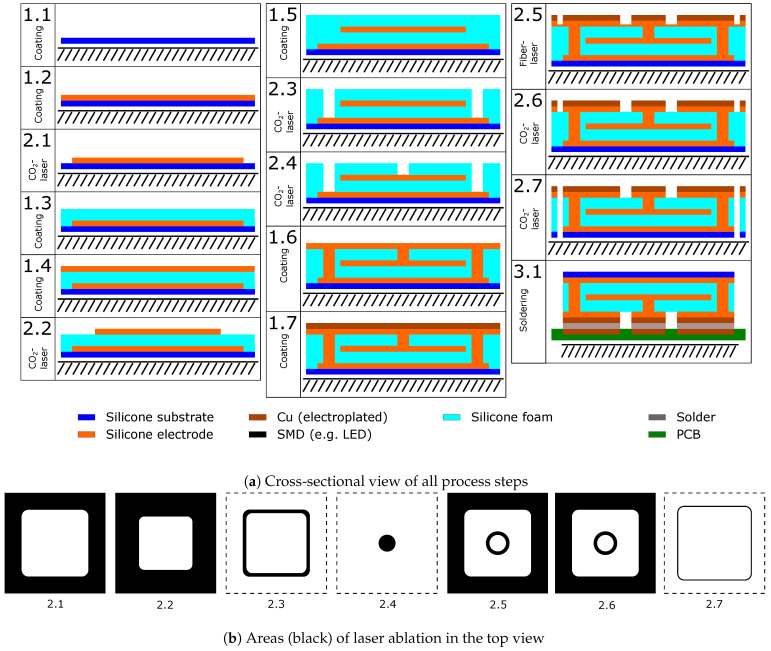
Schematic representation of the coating and laser structuring steps for manufacturing a dielectric elastomer pressure sensor (drawing not to scale).

**Figure 4 micromachines-12-00255-f004:**
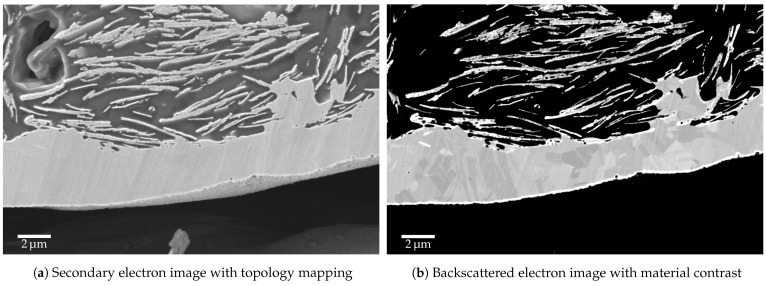
SEM images of the electroplated silicone electrode: particle-silicone composite with gas entrapment (top), metallic copper layer (center), embedding agent (epoxy resin, bottom).

**Figure 5 micromachines-12-00255-f005:**
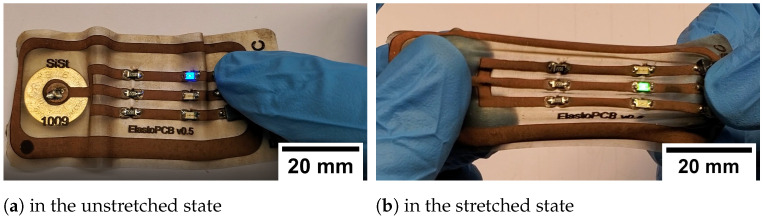
Manufactured circuit board equipped with electrical components in the unstretched state with activated blue LED (**a**) and in the stretched state with activated green LED (**b**). To turn on the LEDs, the respective switches were pressed with the thumb.

**Figure 6 micromachines-12-00255-f006:**
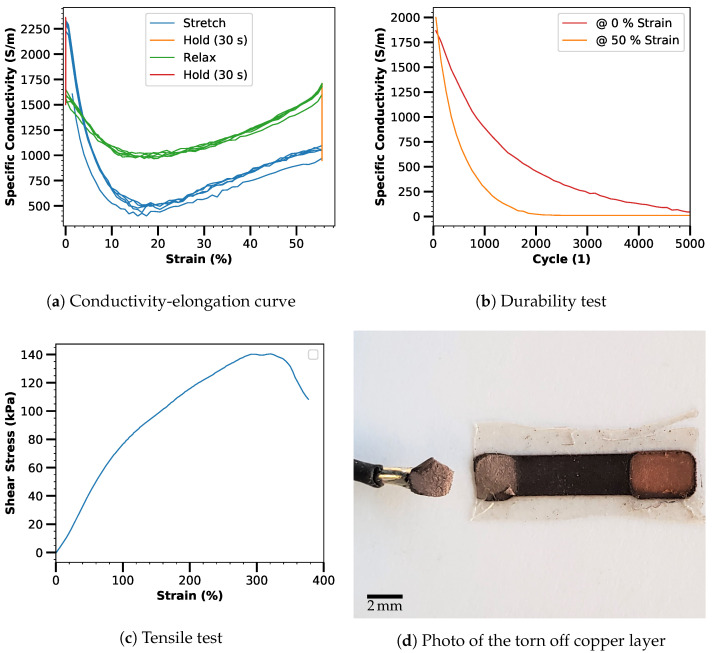
Electromechanical characterization of the laser-fabricated elastic conductor path: Strain dependence of conductivity in five consecutive cycles (**a**), conductivity in the stretched and unstretched states over the span of 5000 strain cycles (**b**), stress-strain diagram to determine the strength of the bonding of the metallic copper layer (**c**), as well as photograph of the (inverted) soldered cable with the copper layer torn off and the residual layer of the silicone composite electrode adhering (**d**). Sharpness and brightness of the photo were adjusted for better recognizability.

**Figure 7 micromachines-12-00255-f007:**
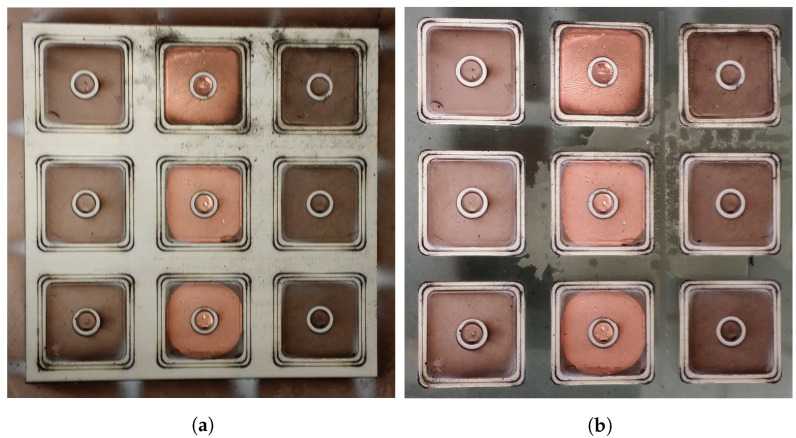
Laser-structured sensors (3 × 3) inside the multi-layer silicone composite film on a glass substrate (**a**) and after removal of the overhanging silicone composite film (**b**). Galvanic metallization took place only for the middle three sensors.

**Figure 8 micromachines-12-00255-f008:**
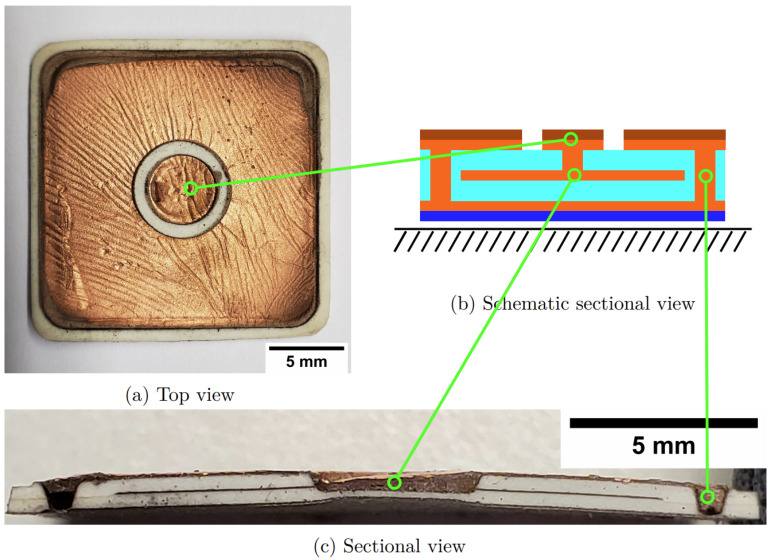
Top view (**a**), schematic cross-sectional view (**b**) and cross-sectional view (**c**) of a manufactured pressure sensor. The insulation ring between the inner and outer electrode is not visible in the cross-sectional view due to the electrode material behind it.

**Figure 9 micromachines-12-00255-f009:**
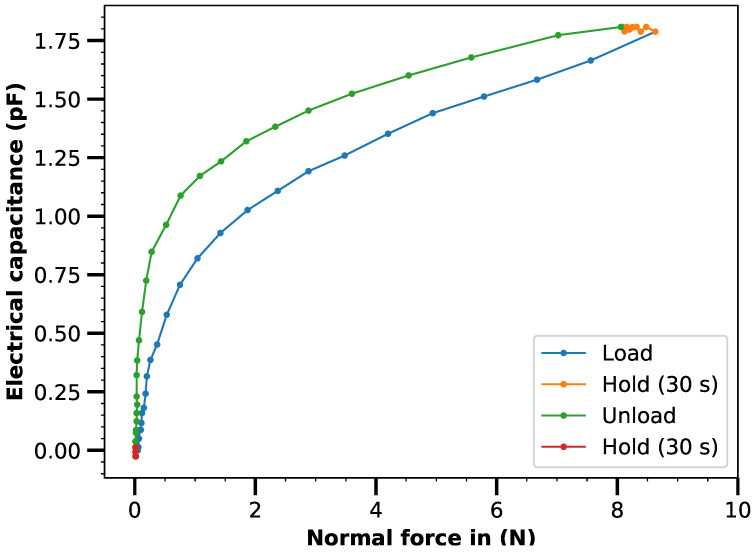
Characteristics of the laser-structured pressure sensor. The absolute change in capacitance compared to the basic capacitance is shown.

**Table 1 micromachines-12-00255-t001:** Silicone formulation *S2* [15].

Component		
Divinyl-terminated PDMS	Content (wt%)	87
	Viscosity (mPa·s)	2000
	c(Vinyl) (mmol g^−1^)	0.09
SiH-terminated PDMS	Content (wt%)	6
	Viscosity (mPa·s)	500
	c(SiH) (mmol g^−1^)	0.16
SiH-Crosslinker	Content (wt%)	7
	Viscosity (mPa·s)	500
	c(SiH) (mmol g^−1^)	1.1

**Table 2 micromachines-12-00255-t002:** Technical data of the used cutting and engraving laser Epilog Fusion Pro 32 [16,17].

Parameter	CO_2_ Laser	Fiber Laser
Wavelength	10,640 nm	1062 nm
Technology	CO_2_ gas tube laser	Ytterbium solid-state laser
Pulse rate	10 Hz–5 kHz	20 kHz–80 kHz
Power	80 W	30 W
Lens	3 Inch	
Focus width ^a^	80 μm (2 Inch Lens)	8.9 μm (5 Inch Lens)
Focus width ^b^	120 μm (3 Inch Lens)	5.3 μm (3 Inch Lens)
Beam profile	Gaussian	Gaussian
Traversing velocity (max.)	4.2 m s^−1^	
Processing velocity (max.)	5 cm^2^s^−1^ or 1.8 m^2^h^−1^ (at 120 μm Focus width)	
Repeatability	0.0127 mm	
Resolution	75–1200 dpi	
Workspace	813 mm × 508 mm	

^a^ Manufacturer specification; ^b^ Calculated according to Equation (3).

**Table 3 micromachines-12-00255-t003:** Laser system parameters for manufacturing the stretchable circuit board.

Step	Description	Mode	Laser	Resolution in dpi	Velocity in % ^a^	Power in % ^a^	Puls Rate in % ^a^	Repetitions
2.1	Remove copper	Engrave	Fiber	1200	100	50	1	1
2.2	Remove copper and electrode	Engrave	Fiber	1200	50	100	1	1
2.3	Cleaning	Engrave	CO_2_	600	100	2	- ^b^	1
2.4	Drilling	Engrave	CO_2_	900	100	10	- ^b^	1
2.5	Cutting out	Cut	CO_2_	-	100	25	100	1

^a^ from the maximum value according to Table 2; ^b^ predefined in CO_2_ engraving mode.

**Table 4 micromachines-12-00255-t004:** Laser manufacturing parameters of the dielectric pressure sensor.

Step	Layer	Mode	Laser	Resolution in dpi	Velocity in % ^a^	Power in % ^a^	Puls Rate in % ^a^	Repetitions
2.1	Electrode 1	Engrave	CO_2_	900	100	1	- ^b^	1
2.2	Electrode 2	Engrave	CO_2_	900	100	1	- ^b^	1
2.3	Dielectric 1 + 2	Engrave	CO_2_	900	100	2	- ^b^	5
2.4	Dielectric 2	Engrave	CO_2_	900	100	1	- ^b^	3
2.5	Electrode 3 + Copper	Engrave	Fiber	-	100	100	50	1
2.6	Electrode 3	Engrave	CO_2_	600	100	2	- ^b^	1
2.7	-All-	Cut	Fiber	-	100	50	100	1

^a^ from the maximum value according to Table 2; ^b^ predefined in CO_2_ engraving mode.

## Data Availability

Data is contained within the article.

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
