# Peer review of "Electroplating and Ablative Laser Structuring of Elastomer Composites for Stretchable Multi-Layer and Multi-Material Electronic and Sensor Systems"

_micromachines, 2021, doi:10.3390/mi12030255_

Round 1
Reviewer 1 Report
Summary: The authors present a new method of creating flexible circuits on soft and porous elastomeric substrates with simple equipment broadly available to the general public. The method of fabrication is new and timely, however some additional experiments are required.
The authors also need to back up claims throughout the manuscript with appropriate citations.
Comment 1: Please add citations to line 45. “The layer applied by vapor deposition is also subject to high equipment costs and adheres only inadequately to the substrate.” Thin film adhesion can be high if engineered correctly. Its not clear what the authors are referring to
Comment 2: In line 58 please define “cycle-resistant stretchability” and clearly define “few percent”. With the described technique stretchability of well over 200% via self-similar serpentines have been demonstrated ( https://doi.org/10.1038/ncomms2553).
Structuring flexible circuit (highly scalable and easily available via commercial flex circuit companies or rapid prototyping via laser ablation) with well-engineered serpentines can achieve also large deformations (https://doi.org/10.1073/pnas.1920073117 figure 3L)
Comment 3: Please add citations to line 73. “The structuring by film masks in combination with doctor blade or spray coating, which is frequently used on a laboratory scale, is very labor-intensive, difficult to automate and offers only low reproducibility.”
Comment 5: Please add citations to line 76. “At the same time, the process is technically limited in terms of minimum structure sizes (not below 0.5mm) and complexity of layer structures (especially with regard to vias).”
Comment 7: The authors are missing crucial mechanical information about the durability of the circuit using this manufacturing method in the result section. Demonstration of cyclic stability of the created devices should be characterized. Especially strain concentrations at the SMD component/soft substrate interface should be documented
Comment 9: What is the durability and reliability of this pressure sensor over many cycles.
Comment 10: What is the reproducibility of such a fabrication for the pressure sensor? What is the standard deviation of capacitance from unloaded and under load between manufactured batches?
Comment 11: Figure 8 shows negative capacitance. Please explain why this is the case.
Comment 12: The authors emphasize the low cost nature of the method however do not present a cost analysis. For example, what cost would PDMS encapsulation of flex circuits with serpentine designs cost in comparison to this method?
Comment 12: Please characterize conductivity of the films (preferably 4 point probe measurement).
Comment 13: What is the minimal trace/space width using this method? For soft electronics a high level of miniaturization is typically required, feature min. feature size is often critical. Also small SMD components such as microcontroller often have 0.4 mm features.
Author Response
- PVD, unlike electrodeposition, cannot fill the undercuts of a porous surface, therefore anchoring of the deposited layer cannot be expected to the same extent.
- in the publication by Schreivogel/Würth Elektronik, who are also working on commercialization, only about 100 cycles to failure were achieved at 20% strain. Even if in the mentioned publication more than 100% was reached for a few cycles, the fundamental problem is the brittleness (in relation and polymers) and the low tear resistance of copper, so that a tearing and thus a total failure can occur at any time during a cyclic load.
- we have marked the statement as our subjective assessment
- <empty>
- see 3.
- <blank>
- we have added a cyclic electro-mechanical test and tensile tests to determine the elongation at break
- <empty>
- the fatigue strength of the sensor is still limited mainly by the electromechanical wear of the particle-silicone composite (see added measurement results) and less by the other components and the structure itself. Moreover, the sensor shown here is only intended to represent the basic function for simple detection tasks.
- we were only able to produce one batch in this series of experiments, and only 3 sensors were metallized, one of which was destroyed for the sectional view. However, the scatter between sensors is not critical, as individual calibration is easy.
- the base capacitance (about 15 pF) was subtracted from the capacitance value for presentation. Due to the measurement uncertainty, values just below zero can therefore also be seen.
- compared to the serpentine conductors mentioned above, the process we have shown has no significant cost advantage, but laser structuring can also be used to process the dielectric layers (such as foam) for sensory applications in the same process. Compared to screen printing, the process we describe has several advantages:
- no tool is required, the manufacture of which always represents a time and financial disadvantage
- the material to be processed is cured before structuring and does not need to be adjusted in terms of its viscosity.
- the achievable resolution should be comparable, but very thin (10 µm) as well as very thick layers can be processed (1 mm)
- (12) We have created a resistance-strain curve of a laser-structured conductor
- (13) The achievable structure sizes are in the range of the focus width of the laser (CO2: approx. 100 µm). 0.2 mm conductors with 0.4 mm are therefore conceivable, but further work is necessary for this.
Reviewer 2 Report
The major novelty of the work lies in fabrication method for multilayered stretchable circuit or sensors with laser engraving. I recommend publication after addressing the following questions:
- The fabrication step seems to be complicated. It involves more steps compared with screen printing intrinsically stretchable materials layer by layer. Could you please explain the advantages over screen printing, in terms of cost, complexity and spatial resolution?
- The authors should perform electrical and mechanical characterization at the soldered connection, where stress build up on between stretchable materials and non-stretchable electronic component (e.g. resistor or ICs)
- The authors should characterize the conductance change versus different strain for the stretchable conductor fabricated by laser.
- Multiple cycles should be tested for the pressure sensor? How the pressure sensor will be influenced if it is under stretch? Any cross-talk?
Author Response
- compared to screen printing, the process we describe has several advantages:
- no tooling is required, the manufacture of which is always a time and financial disadvantage.
- the material to be processed is cured before structuring and does not need to be adjusted in terms of viscosity.
- the achievable resolution should be comparable, but both very thin (10 µm) and very thick layers can be processed (1 mm)
- we have added a cyclic electro-mechanical test and tensile tests to determine the elongation at break
- we have created a resistance-elongation curve of a laser-structured conductor
- elongation of the shown simple sensor is not possible, because the contacting side was metallized. However, in future development, the contacting side should be spatially separated from the sensor surface to allow strain or strain sensors. In addition, the sensor shown here is only intended to represent the basic function for simple detection tasks.
Round 2
Reviewer 1 Report
The manuscript can be accepted in its current form. However I would encourage the authors to rephrase or remove the section about current techniques that use flex circuits or clean room fabricated devices. There are several reports from the Rogers group an others that demonstrate cyclic strain stability and operation in highly demanding applications such as implants (one example here https://doi.org/10.1038/s41467-019-13637-w).
I think the argument that this technique can be done in labs with minimal equipment is sufficient for the motivation of this work.
Author Response
We rephrase the section
Reviewer 2 Report
The authors addressed most of the questions and improved the manuscript.
Author Response
(no further comments)